# Surgical Treatment of Lithiasis of the Main Pancreatic Duct: A Challenging Case and a Literature Review

**DOI:** 10.3390/diseases12050086

**Published:** 2024-04-30

**Authors:** Dan Brebu, Cătălin Prodan-Bărbulescu, Vlad Braicu, Paul Pașca, George Borcean, Sabrina Florea, Clarisa Bîrlog, Amadeus Dobrescu, Mărioara Cornianu, Fulger Lazăr, Bogdan Totolici, Ciprian Duță, Flaviu Ionuț Faur

**Affiliations:** 12nd Surgery Clinic, Timișoara Emergency County Hospital, 300723 Timisoara, Romania; brebu.dan@umft.ro (D.B.); braicu.vlad@umft.ro (V.B.); paul.pasca@umft.ro (P.P.); borceang@yahoo.com (G.B.); amadeusdobrescu@umft.ro (A.D.); lazarfulger@yahoo.com (F.L.); duta.ciprian@umft.ro (C.D.); flaviu.faur@umft.ro (F.I.F.); 2X Department of General Surgery, “Victor Babeș” University of Medicine and Pharmacy, 300041 Timisoara, Romania; 3Department I, Discipline of Anatomy and Embriology, “Victor Babeș” University of Medicine and Pharmacy, 300041, Timisoara, Romania; 4Doctoral School, “Victor Babes” University of Medicine and Pharmacy Timisoara, Eftimie Murgu Square 2, 300041 Timisoara, Romania; 5Department of Medical—Clinical Disciplines, Faculty of Medicine, “Titu Maiorescu” University of Bucharest, 031593 Bucharest, Romania; 6Medicine Doctoral School, “Titu Maiorescu” University of Bucharest, 031593 Bucharest, Romania; 7Department of General Surgery, Monza Clinical Hospital, 021967 Bucharest, Romania; 8Department of General Surgery, Sanador Clinical Hospital, 010991 Bucharest, Romania; 9Department of Surgery, Ponderas Academic Hospital, 021188 Bucharest, Romania; clarisa.birlog@reginamaria.ro; 10Department of Microscopic Morphology-Morphopatology, ANAPATMOL Research Center, “Victor Babes” University of Medicine and Pharmacy, 300041 Timisoara, Romania; 11Department of Pathology, “Pius Brinzeu” County Clinical Emergency Hospital, 300723 Timisoara, Romania; 121st Clinic of General Surgery, Arad County Emergency Clinical Hospital, 310158 Arad, Romania; totolici_bogdan@yahoo.com; 13Department of General Surgery, Faculty of Medicine, “Vasile Goldis” Western University of Arad, 310025 Arad, Romania; 14Multidisciplinary Doctoral School “Vasile Goldis”, Western University of Arad, 310025 Arad, Romania

**Keywords:** pancreaticolithiasis, Wirsung lithiasis, pancreas duct stone, lithiasis of main pancreatic duct, pancreatic calculi, corporeo-caudal pancreatectomy, Kimura procedure

## Abstract

Pancreaticolithiasis represents a rare phenomenon, being superimposed most of the time on a form of chronic pancreatitis of multifactorial etiology. Pancreaticolithiasis is a late complication of the phenomenon of chronic pancreatitis. The reverberant inflammatory process, followed by the fibrotic degeneration of the pancreatic parenchyma, and pancreatic fluid stasis at the ductal level are factors that contribute to the phenomenon of calcium precipitation. This article describes the case of a patient with a diagnosis of pancreaticolithiasis (Wirsung duct lithiasis), a phenomenon superimposed on chronic pancreatitis of ethanolic cause (Rosemont classification). It was decided to perform surgery via the classical approach with the perfection of corporeo-caudal pancreatectomy and preservation of the splenic vessels (Kimura procedure) with pancreatico-jejunal anastomosis on the Roux-en-Y loop. The aim of this study is to identify the best method of treatment for pancreaticolithiasis. To enhance the case and provide a basis for standardization, a literature review was carried out, which included a total of six articles. The results of this study highlight that, currently, the management of symptomatic pancreaticolithiasis encompasses medical therapy (enzyme replacement therapy), interventional therapy (ESWL (extracorporeal shock wave lithotripsy) ± ERCP (endoscopic retrograde cholangiopancreatography), ERCP + sphincterotomy + stent insertion, and POP (oral pancreatoscopy)), and surgical treatment. In conclusion, based on the analysis conducted in this study, the size of the calculi present determines which is the suitable therapeutic care. Unlike stones over 0.5 cm, when surgery is explicitly advised for therapeutic purposes in the absence of endoscopic techniques, stones under 0.5 cm should be treated using endoscopic procedures.

## 1. Introduction

The pancreas is a soft, lobulated, and elongated exo-endocrine gland located at the level of the epigastric and left hypochondriac regions of the posterior abdominal wall. It is extended between the duodenum and spleen, opposite the level of the T12-L3 vertebrae. A significant portion of the gland is retroperitoneal, behind the serous surface of the smaller sac. The left extremity’s tail is located in the lienorenal ligament, indicating that it is an intraperitoneal component of the pancreas [1,2]. The pancreas is divided into four sections: the head, neck, body, and tail. The superior mesenteric vein marks the junction between the head and the isthmus of the gland. The pancreatic head has a posterior and inferior lengthening called the uncinate process. The section above the superior mesenteric vein is known as the istm, and the body extends to its left. The tail runs in the direction of the spleen’s hilum.

The pancreas is a highly vascular structure with arterial sources, including the celiac trunk and superior mesenteric artery. These vessels provide oxygenated blood supply to the pancreas. According to Nomica Anatomica, at the level of the pancreatic regions there are the following arteries: the superior and inferior pancreaticoduodenal arteries, which are divided into anterior and posterior branches. These branches form the anterior and posterior pancreaticoduodenal arterial arcades. Anastomoses ensure oxygenated blood supply for the pancreatic parenchyma. Pancreatic veins drain into the splenic, superior mesenteric, and portal veins. Pancreaticosplenic lymph nodes receive lymph from the body and tail of the pancreas through lymphatic vessels. The lymph is then moved to the celiac or superior mesenteric lymph nodes [1,2].

The pancreas is composed of lobules with connective tissue between them, which are connected by excretory tubules. The pancreatic tissue has two functions: exocrine and endocrine. The lobules have acinous tissue (alveolar) with external secretion, and insular tissue (islets of Langerhans) with internal secretion. Endocrine cells form a network of blood capillaries, and the islets of Langerhans are visible macroscopically. The alveolar pancreas secretes a preferment that converts enterokinase into trypsin, while the exocrine pancreas pours secretion products into the duodenum II [1,2].

The pancreatic ductal system consists of interlobular ducts, main pancreatic ducts, accessory pancreatic ducts, and intralobular ducts that connect acinar tubules. These components are mainly visible through light and electron microscopy. The duct system is crucial for maintaining the integrity of the pancreas, as exocrine enzymes can cause tissue damage and lead to pancreatitis. The pancreatic self-digestion model shows that the consistency of the pancreatic duct walls is related to collagen content. As ducts branch, the connective tissue becomes thinner, and tight intercellular connections—known as zonula occludens—link ductal cells, centroacinar cells, and acinar cells, limiting leakage from the ductal system [1,2].

In terms of morphopathology, the inflammation of the pancreas is known as acute pancreatitis and chronic pancreatitis. Since self-digestion and chronic inflammation cause the pancreas to fibrose and atrophy, acute pancreatitis and chronic pancreatitis are multifactorial, systemic diseases that can progress from one to the other [3,4].

Chronic pancreatitis is a progressive condition characterized by irreversible structural changes affecting endocrine and exocrine pancreatic function. The main cause is chronic alcoholism, accounting for 75% of etiopathogenesis. According to Figure 1, the etiology includes genetic factors, congenital anomalies, systemic metabolic disorders, biliary disorders, acute pancreatitis, and chronic idiopathic pancreatitis. Other contributing factors include mutations in the cystic fibrosis transmembrane conductance regulator, PRSS1 gene, and Kazal type 1 gene [3,4,5].

Pancreaticolithiasis is a condition that can occur in the pancreatic ducts (Wirsung or Santorini), side branches, or in the pancreatic parenchyma. From a compositional (chemical) point of view, pancreatic calculi contain an inner nidus surrounded by successive layers of calcium carbonate [3,4,5].

Pancreaticolithiasis represents a rare phenomenon, being superimposed most of the time on a form of chronic pancreatitis of multifactorial etiology. Pancreaticolithiasis is a late complication of the phenomenon of chronic pancreatitis. The reverberant inflammatory process, followed by the fibrotic degeneration of the pancreatic parenchyma, and pancreatic fluid stasis at the ductal level are factors that contribute to the phenomenon of calcium precipitation [3,4,5].

Calcium is present in a significant amount in the pancreatic juice, which is kept within physiological limits by the actions of HCO^3−^, citrate, and pancreatic stone protein (PSP). Against the background of parenchymal destruction occurring in chronic pancreatitis, the regulatory mechanisms are compromised, a phenomenon that results in the appearance of pancreaticolithiasis. The occurrence of calculi at the level of the Wirsung canal has as a consequence the disruption of pancreatic outflow, the appearance of ductal hyperpressure, and finally ischemic degeneration [3,4,5].

Regarding the establishment of the diagnosis of pancreaticolithiasis or lithiasis of the Wirsung duct, endoscopic ultrasonography (EUS) and magnetic resonance cholangiopancreatography (MRCP) show a high sensitivity.

Referring to therapeutic aspects, endoscopic retrograde cholangiopancreatography (ERCP) with endoscopic sphincterotomy (ES) and with extraction, sometimes in combination with extracorporeal shock wave lithotripsy (ESWL), represents the treatment of choice in the case of pancreaticolithiasis, but this is not a gold standard due to frequent recurrence [6].

It is considered that endoscopic ultrasonography (EUS) is the most sensitive method for identifying chronic pancreatitis. Eleven criteria that indicate parenchymal and ductal features are used to classify chronic pancreatitis [7].

The Rosemont classification is a useful instrument for standardizing the endoscopic diagnosis of chronic pancreatitis. This classification uses established endoscopic ultrasonography (EUS) parameters to group individuals receiving endosonography according to their risk of developing chronic pancreatitis. It is unknown how useful these criteria will be in predicting the prognosis and treatment outcomes of patients with chronic pancreatitis. This study’s hypothesis was that if patients fit the Rosemont criteria for chronic pancreatitis at the time of EUS and had stomach discomfort and clinical concern about the condition, they would be more likely to experience a pain response to pancreatic enzyme supplementation. Additionally, we looked for the endosonographic criteria for chronic pancreatitis that would most accurately indicate a decrease in pain following pancreatic enzyme supplementation therapy [7].

Pancreaticolithiasis is classified according to the type of stones found, their number, and their location. From an imaging point of view, there are radiopaque, radiotransparent, and mixed-structure stones, with a significant frequency of radiopaque ones. Referring to their number, they can be singular or multiple, and from the point of view of the location, we can identify calculi located at the level of the main pancreatic duct or at the level of the accessory ducts. From a topographic point of view, pancreaticolithiasis is evident at the cephalopancreatic level or at the corporeo-caudal level [3,4,5].

## 2. Case Report

A 60-year-old patient, known to be a smoker, was referred to the emergency department complaining of moderate abdominal pain located in the epigastric region with posterior irradiation, accompanied by dyspepsia.

Regarding his medical history, he was reported as suffering from chronic ethanol pancreatitis (Rosemont classification) and a perforated gastric ulcer for which a Pean-Billroth I gastrectomy was performed.

The usual blood tests identified the following: leukocytosis (15,700/mm^3^) glucose level = 117 mg/dL; ALT, AST and LDH within normal limits; amylase = 153 U/L; lipase = 160 U/L; cholinesterase = 5822 U/L.

The imaging assessment revealed the following:Abdominal ultrasonography: Liver with homogeneous structure, without intrahepatic bile duct (IHBD) dilatations, distended gallbladder 10.8/3.2 cm, without stones, common bile duct (CBD) of approximately 7 mm. Pancreas with Wirsung duct visibly dilated up to 13 mm with multiple hyperechoic images, and cephalic pancreatic region increased with inhomogeneous structure and multiple calcifications. Multiple infracentimeter transonic formations at the level of the cephalopancreatic region.Magnetic resonance cholangiopancreatography (Cholangio-MR): Destructuralized, inhomogeneous pancreas with fine inflammatory changes at the cephalopancreatic level; retrograde dilation of the Wirsung canal up to 15 mm above the isthmic segment with inhomogeneous content, and several stones of 1 cm diameter with obstructive pattern. Without dilatation of intrahepatic bile ducts. No hypo/hyperabsorbing areas suggestive of malignancy (Figure 2).

Corroborating clinical and paraclinical aspects, the diagnosis of acute pancreatitis of ethanolic cause was established in mild form (Ranson 1, Atlanta classification), with Wirsung duct lithiasis; the phenomenon was superimposed on chronic pancreatitis of ethanolic cause (Rosemont classification) [7].

## 3. Case Outcome

### 3.1. Treatment Plan

The patient was dispensed in the Gastroenterology and Hepatology Department for metabolic balancing. Therapeutic aspects regarding the management of pancreaticolithiasis were discussed among the multidisciplinary team (BPH surgeon, gastroenterologist, diabetologist, radiologist, cardiologist, anesthesiologist).

### 3.2. Surgical Intervention

It was decided to perform surgery via the classical approach with the perfection of corporeo-caudal pancreatectomy and the preservation of the splenic vessels (Kimura procedure) with pancreatico-jejunal anastomosis on the Roux-en-Y loop [8]. 

The indication for surgery is based on specialist guidelines that are in line with the size of the stones. In this case, as there were several stones with a size exceeding 0.5 cm, surgery was indicated.

### 3.3. Operative Steps

Surgical intervention: Corporeo-caudal pancreatectomy with preservation of the splenic vessels (Kimura procedure) and pancreatico-jejunal anastomosis on the Roux-en-Y loop (Figure 3).

### 3.4. Actual Outcome

The postoperative outcome was favorable. The patient was transferred to the Diabetology, Nutrition and Metabolic Diseases Clinic on postoperative day 10 to evaluate the status of postresectional pancreatic function and metabolic compensation.

During hospitalization the patient was monitored clinically and paraclinically with investigation into a possible diagnosis of postresectional diabetes mellitus; thus, HbA1c was performed, showing a value of 6.8%, suggesting a diagnosis of diabetes mellitus pre-existing the corporeo-caudal pancreatectomy. Subsequently, for the evaluation of the remaining endocrine pancreatic secretion, C-peptide was performed, which showed a value at the lower limit of normal, namely 0.97 ng/mL.

Considering that during the hospitalization the patient showed blood glucose values within normal limits, it was decided that blood glucose values could be self-monitored at home with a re-evaluation 10 days after discharge with therapeutic reconfiguration. Pancreatic enzyme replacement treatment was continued with pancrelipase at 35,000 units, preprandial.

### 3.5. Follow-Up

In this case, a clinical–biological imaging evaluation was performed at 3 months after discharge, at 6 months, and then at 1 year. The post-treatment evolution was favorable, characterized by the remission of symptoms.

One year after discharge, the patient presented to the Gastroenterology Department with abdominal pain in the epigastric area, nausea, vomiting, and increased pancreatic enzymes (lipase and amylase), on which occasion it was decided to admit him to the Gastroenterology and Hepatology Clinic. The diagnosis of acute pancreatitis was established and specialized medical and supportive treatment was initiated. The patient showed remission of the pancreatic reaction under treatment and was discharged.

Another important factor in the context of follow-up is smoking. The patient received professional anti-smoking counselling to stop this known risk factor. Following counselling, the patient quit smoking.

## 4. Discussion

The occurrence of pancreaticolithiasis is more often associated with chronic pancreatitis than with acute pancreatitis [2]. Pancreaticolithiasis occurs most often in males and is associated with pancreatitis of ethanolic cause [3]. The most important clinical elements for diagnosis are epigastralgia, fever, nausea, vomiting, sweating, and steatorrhea [2].

The formation of pancreaticolithiasis is a complex process involving chronic pancreatitis and biliary tract disease. Pancreatic calculi, primarily composed of calcium carbonate, are formed when pancreatic secretion is altered due to stasis or infection [4]. These stones are typically found in the main pancreatic Wirsung duct, branches, or glandular parenchyma, and are typically less than 1 cm in diameter and sandy in shape. They have a high calcium content in the form of carbonate and phosphate, resulting in their high density [2].

Pancreaticolithiasis can be diagnosed paraclinically using plain radiography, abdominal ultrasound, endoscopic ultrasound, endoscopic retrograde cholangiopancreatography (ERCP), cholangiopancreatography magnetic resonance (MRCP), CT scan, and MRI [9].

Endoscopic retrograde cholangiopancreatography (ERCP) is the most precise method for determining ductal dilatation and intraductal lithiasis [8]. The purpose of these radiological examinations is to classify pancreatic lithiasis based on its type, number, and location. They can be radiopaque, radiolucent, or mixed; single or multiple; located in the main pancreatic duct, side branches, or pancreatic parenchyma; and in the head, body, or tail regions [9].

Pancreaticolithiasis can be detected using plain abdominal radiographs, which show the location and appearance of calculi. Abdominal ultrasound, due to its high reflectivity, can detect intraductal calculi, allowing for the study of the main pancreatic duct’s state and the selection of patients for surgery, especially when ERCP has not been successful [9].

Pancreaticolithiasis can be detected in MR cholangiopancreatography as a low-signal foci within the pancreatic duct, which may be surrounded by a high signal of pancreatic fluid (meniscus sign) [9].

### Literature Review

A brief advanced PubMed search on the topic—conducted using the MeSH terms “pancreatolithiasis” and “Wirsung lithiasis”—yielded a total of 22 results by filtering the study level to original articles. The total number of articles included in this study is six (Figure 4 and Table 1).

Analyzing the specialized literature, we can summarize some objective aspects regarding the management of pancreaticolithiasis. Currently, the management of symptomatic pancreaticolithiasis encompasses medical therapy (enzyme replacement therapy), interventional therapy (ESWL (extracorporeal shock wave lithotripshy) ± ERCP (endoscopic retrograde cholangiopancreatography), ERCP + sphincterotomy + insertion of a stent, or POP (oral pancreatoscopy), and surgical treatment.

From a surgical point of view, corporal-caudal pancreatectomy is one of the most common surgical interventions for lesions affecting the body and tail of the pancreas. It is accepted, according to the literature, that for non-malignant pathologies, corporeo-caudal pancreatectomy with preservation of the spleen is the surgery of first choice.

Splenectomy is required when there is a malignant pancreatic lesion to perform a lymphadenectomy that includes the lymph nodes located in the spleen hilum (station 10). It is also well known that the introduction of splenectomy in the surgical arsenal comes with possible post-splenectomy complications, the most relevant of which are infectious complications (intra-abdominal abscesses), hematological complications (venous thrombosis and arterial thrombosis), and pulmonary hypertension. Thus, the preservation of the spleen goes in tandem with the preservation of immune system function [15,16].

Also, an important aspect to consider when performing a spleen-preserving pancreatectomy is the preoperative assessment of the vascularization of the spleen. Surgically, there are two approaches that allow splenic preservation: pancreatectomy with splenic vessel preservation (Kimura technique), or pancreatectomy with splenic vessel resection (Warshaw technique). The Kimura technique seems to be more beneficial, as it avoids the splenic infarctions associated with the Warshaw procedure (Figure 5) [15,17].

From a technical point of view, it is obvious that performing a pancreatectomy with preservation of the splenic vessels is much more laborious and difficult, as it requires the posterior release of the pancreatic tissue to which these vessels are adherent.

The surgical intervention begins with a careful evaluation of the greater peritoneal cavity and intra-abdominal organs. The dissection begins by accessing the omental bursa through the gastrocolic ligament with its dissection from the midline to the left, following the avascular plane until the identification of short gastric vessels with their preservation and the arch of the great gastric curvature.

The spleno-colic ligament is dissected with mobilization of the splenic flexure of the colon to expose the pancreatic tail. The stomach is retracted cranially; this is called “cephalisation of the stomach”. The lower pancreatic border is identified (the location from which the dissection is started), followed by moving towards the posterior aspect of the pancreas to identify the splenic vein, which is carefully dissected and released. A technical variant is to identify per primam the splenic artery with its ligation. From a hemodynamic point of view, after clamping the splenic artery, splenic and pancreatic flow will be reduced and blood will return to the splenic vein, highlighting that the splenic vein is easier for dissection.

Once the pancreatic corporeo-caudal portion has been mobilized, the decision is made to interrupt the pancreatic parenchyma at the level of the isthmic area. The pancreatic stump is examined and hemostasis is performed. Spleen perfusion is also assessed. Pancreatico-jejunal Roux-en-Y anastomosis is performed.

Pancreatico-jejunal anastomosis is a demanding procedure in surgery; it is to be performed following a pancreatectomy (cephalic or corporeo-caudal) and aims to drain pancreatic juice from the remaining pancreatic tissue in the gastrointestinal tract. Various methods have been attempted to close the main pancreatic duct, but these methods have been clinically ineffective due to the risk of postoperative pancreatitis. Total pancreatectomy is another option, especially for high-risk patients with soft pancreatic tissue and a major pancreatic duct < 3 mm. The auto-transplantation of pancreatic islets is also proposed to prevent postoperative diabetes management. Regardless of the risk of postoperative pancreatic fistula, an anastomosis between the pancreatic stump and gastrointestinal tract remains the most effective and safe method for securing the remaining pancreas [18].

The evolution of patients with pancreaticolithiasis is dependent on the multimodal treatment (medical treatment, interventional, and/or surgical). In the absence of treatment, a patient with diagnosed pancreatic lithiasis evolves complications. Complications of pancreaticolithiasis include the following:Local complications (complications involving the biliopancreatic complex)

Pancreaticolithiasis can lead to the following local complications: recurrent pancreatitis; cholecystitis and cholangitis; diabetes. Pancreatic lithiasis is responsible for the occurrence of pancreatic pseudocysts, necrotizing pancreatitis, and pancreatic cancer. This is a rare complication of pancreatic stones. Pancreatic cancer is more likely to develop in patients with long-standing pancreatitis [5].

B.Systemic complications

Pancreaticolithiasis can lead to the following systemic complications: peritonitis, hypovolemic shock, septic shock (could arise from the small intestine’s bacteria backwashing into the ducts as a result of the stone blockage; this infection may occasionally enter the bloodstream and result in sepsis, a potentially fatal condition), and malnutrition (pancreatic stones may cause problems with digestion and nutrient absorption; this illness may cause starvation and weight loss) [5,8].

The treatment of pancreaticolithiasis depends on the location, size, and number of stones. The European Society of Gastrointestinal Endoscopy recommends ERCP for small stones below 5 mm in the proximal part and body of the pancreas. If the stones block the main pancreatic duct, ESWL is recommended before ERCP [16].

According to the study conducted by Li et al., B-ultrasonography is the preferred method for identifying pancreatic lithiasis, while MRCP offers higher specificity in diagnosing and treating pancreaticolithiasis. According to Li et al. Surgical therapy is the curative method for resolving pancreaticolithiasis with favorable long-term results [10].

Saghir SM and colleagues initiated a review that included a number of 361 studies regarding the management of pancreaticolithiasis, and a number of 16 studies reporting the beneficial role of per oral pancreatoscopy (POP) in the case of pancreaticolithiasis. POP-guided lithotripsy is a feasible therapeutic option, but it must be systematized according to clinical characteristics, the logistics of the respective interventional team, and experience. The need for randomized trials is imperative in order to evaluate the possibility of implementing POP as the first therapeutic line in selected cases of pancreaticolithiasis [17].

According to the 2018 European Society of Gastrointestinal Endoscopy (ESGE) and the 2017 United European Gastroenterology (UEG) guidelines, ESWL is the first-line therapy for patients with symptomatic pancreaticolithiasis unresponsive to medical therapy and presenting stones with a diameter over 5 mm. ERCP remains a feasible procedure in the case of stones with a diameter of less than 5 mm [18].

Gutierrez et al. report in their study that the presence of three or more calculi in the pancreatic duct represents an independent risk of failure of POP (*p* ˂ 0.04) [17].

Kondo et al. found that pancreatic duct stenting before ESWL reduced the shock waves required for stone fragmentation and shortened therapy duration. If unsuccessful, patients may be eligible for surgery, with the Partington–Rochelle modification of the Puestow procedure being the most common surgical technique [19].

Pancreaticolithiasis recurrence rates are higher in patients with main pancreatic duct stenosis (50 vs. 13%), and 22% for ERCP combined with ESWL, while post-procedural pain is less frequent after surgery compared to endoscopic treatment [16,18].

According to studies by Roberto L Meniconi et al., it should be pointed out that, in the case of multiple Wirsung stones, they are often embedded in the canal and are difficult to remove because of their angular shape. In these cases, digestive anastomosis must include an incision of the parenchyma with a flattening of the dilated Wirsung duct and the creation of Puestow’s lateral pancreato-jejunostomy, modified by Partington and Rochelle [19].

In cases where pancreatic lithiasis is characterized by small and easily removable stones, the distal resection of the pancreas with head-to-head pancreato-jejunostomy is the best solution compared to the closure of the remaining pancreatic stump [19].

Due to the relatively low incidence of this pathology, there are few patients with primary pancreatic lithiasis, which correlates with relatively limited medical and surgical expertise. This is the reason why guidelines are not unanimously accepted and there is still no standardization of the therapeutic steps that need to be implemented.

In terms of future research directions, there are a couple of particularly relevant considerations:The early identification of risk factors and the establishment of a patient screening system.The standardization of an accurate diagnostic method in pancreaticolithiasis. Ideally, this method should have the highest sensitivity and specificity and the lowest risk.To identify and implement an effective treatment for patients with pancreaticolithiasis.The stratification of patients using an algorithm and the application of a step-up personalized treatment.Encouraging reporting in the registries of each center and country and publication in the literature via articles, which brings with it the dissemination of expert information and adds value to the management of patients with pancreaticolithiasis.

## 5. Conclusions

In conclusion, the management of pancreaticolithiasis patients does not currently “enjoy” a standard therapeutic protocol, as there is not yet a worldwide consensus.

According to what has been analyzed in this study, it can be stated that therapeutic management is chosen according to the size of the calculi present. In the case of a stone below 0.5 cm, endoscopic treatment is indicated, unlike stones above 0.5 cm, in which case surgery is directly indicated for therapeutic purposes in the absence of endoscopic procedures.

## Figures and Tables

**Figure 1 diseases-12-00086-f001:**
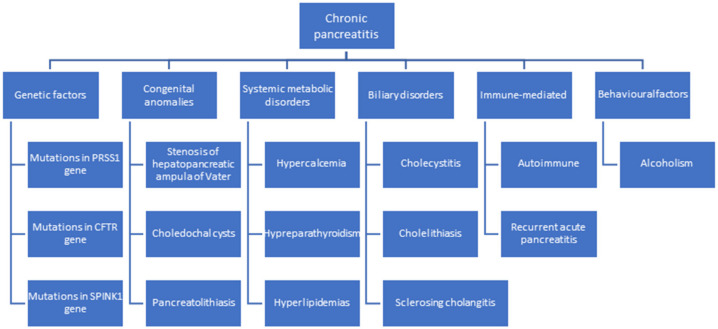
Etiopathogenesis of chronic pancreatitis, highlighting pancreatic lithiasis as a risk factor for chronic pancreatitis.

**Figure 2 diseases-12-00086-f002:**
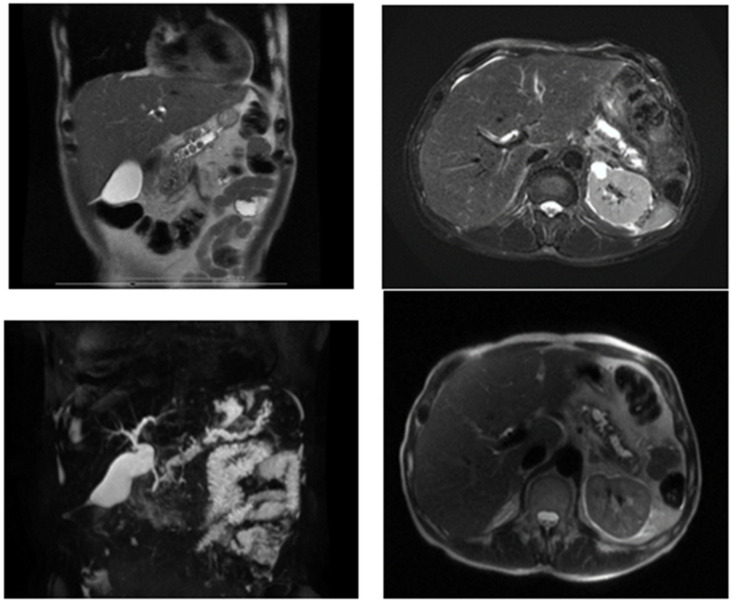
Magnetic resonance cholangiopancreatography.

**Figure 3 diseases-12-00086-f003:**
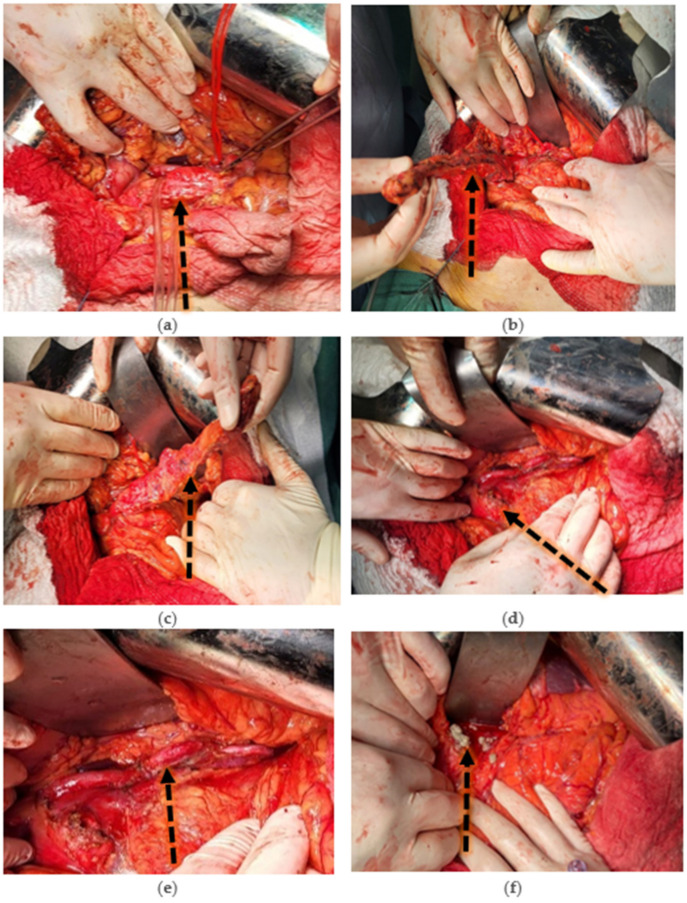
(**a**) Exposure of the pancreatic lodge. Vascular time begins with release of the pancreas by advanced vascular sealing of splenic artery branches (dorsal pancreatic artery, left branch of greater pancreatic artery, inferior pancreatic artery, and artery of the tail of the pancreas) and splenic vein tributaries. (**b**) Release of the pancreas from its means of attachment (Treitz’s retro-pancreatic fascia) with a clear view of splenic artery, splenic vein, and venous splenomesenteric trunk. (**c**) The body and tail of the pancreas are completely released. Preparation of the pancreatic isthmic section is performed. (**d**) Pancreatic section is performed and the remaining cephalo-pancreas is visualized. Splenic artery, splenic vein, and splenomesenteric venous trunk are visualized. (**e**) Corporeo-caudal pancreatectomy with preservation of splenic vessels (Kimura Procedure). (**f**) Evacuation of stones (calculi) from the cephalopancreatic portion of the main pancreatic duct (Wirsung duct).

**Figure 4 diseases-12-00086-f004:**
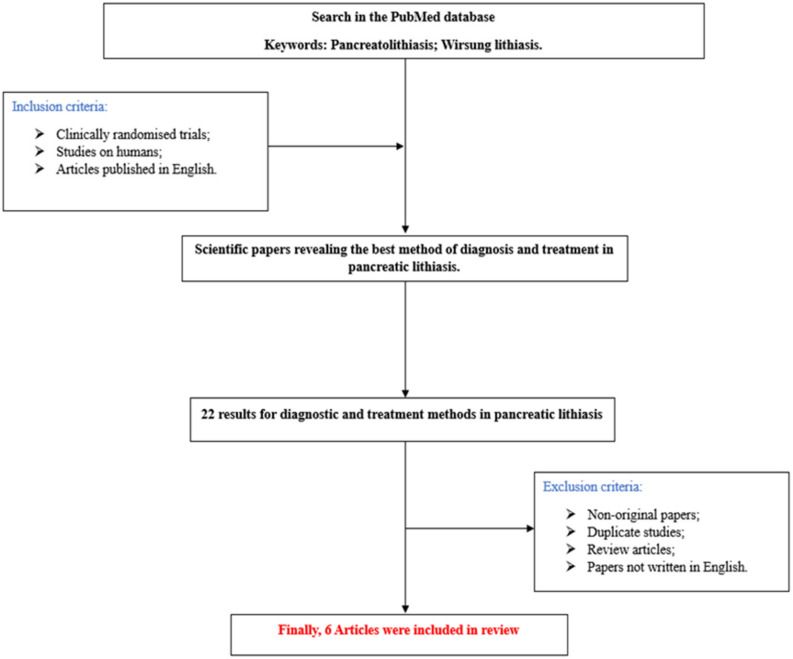
Flowchart of the study.

**Figure 5 diseases-12-00086-f005:**
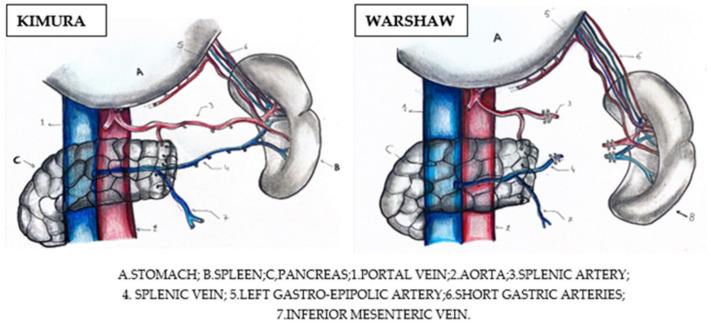
Pancreatectomy with splenic vessel preservation (Kimura) versus pancreatectomy with splenic vessel resection (Warshaw) [15].

**Table 1 diseases-12-00086-t001:** Literature studies identified after advanced research on PubMed (MeSH terms “Pancreatolithiasis” and “Wirsung lithiasis”).

Study Number	Reference	Study Design	Total Number of Patients	Therapeutic Method
1	Li JS [10]	Retrospective study	88	Surgery
2	Díte P [11]	Prospective randomized trial	72	Surgery
3	Cahen DL [12]	Randomized trial	39	Surgical drainage of the pancreatic duct
4	Rutter K [13]	Retrospective study	292	Surgery
5	Cahen [14]	Randomized trial	39	Surgery
6	Van der Hul RL [15]	Original article	16	Extracorporeal shock wave lithotripsy (ESWL)

## Data Availability

Data are contained within the article.

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
