# Peer review of "Surgical Treatment of Lithiasis of the Main Pancreatic Duct: A Challenging Case and a Literature Review"

_diseases, 2024, doi:10.3390/diseases12050086_

Round 1

Reviewer 1 Report

Comments and Suggestions for Authors

This case report provides valuable insights into the challenging presentation of pancreatolithiasis in the main pancreatic duct. The authors effectively navigate through a complex clinical scenario with detailed clinical and surgical images, offering thorough analysis and discussion supported by a comprehensive review of existing literature. The inclusion of a literature review enhances the understanding of this rare condition and underscores the significance of the reported case. Please find my suggestions to improve the current paper noted below.

1.       Different terminology used to describe pancreatolithiasis throughout the paper, including pancreaticolithiasis, pancreatic lithiasis, pancreatic stones etc. Please be consistent.

2.       Title: Lithiasis implies multiple already, multiple can be removed or can switch to Multiple lithiases. Also, after pancratic duct, there should be colon, not period.

3.       Keywords: can be enhanced adding pancreas duct stone, lithiasis, pancreatic calculi

4.       Materials and Methods title should be corrected, as this is a case report; Detailed Case Description, Case Report or Case Presentation or something similar can be used based on editorial preference..

5.       Results: Again, results are not a good title; can use Case outcome etc.

6.       Line 269: Please elaborate why surgical option was favorable so readers would know your thought process

7.       Line 281: Please use generic name such as pancrelipase instead Pankreal.

8.       Line 293: what treatment, supportive care or additional endoscopic or surgical?

9.       Line 293: did patient stop smoking? or counseled? Smoking is one of the known factor, please discuss.

10.   Line 495, 496, where does the below 1cm cut off come from, cited article refers as 0.5cm (reference 21). Please clarify.

Author Response

Please see the attachment below.

Reviewer 2 Report

Comments and Suggestions for Authors

The article highlights a technical problem to be adopted in case of lithiasis of the Wirsung duct. It must be kept in mind that in case of multiple Wirsung stones these are often embedded in the duct and can be removed with difficulty, due to their angular shape. In these cases the digestive anastomosis must include a large incision of the parenchyma with flattening of the dilated Wirsung duct and the creation of a side to side pancreatojejunostomy by Puestow, modified by Partington and Rochelle . Instead, when the stones are small and easily removable, distal resection of the pancreas with pancreato-jejunostmy end to side is the best solution compared to closure of the remenant pancreatic stump. In this regard, an article published in 2013 by our working group is attached.

In conclusion, the article may be published with minor revisions and updating of the bibliography

Author Response

Please see the attachment below.

Reviewer 3 Report

Comments and Suggestions for Authors

l   The manuscript included a case report of PD lithiasis and the related literature review. However, the use of the 'introduction/M&M/results/discussion' format typical of original articles may not be appropriate for this manuscript.

l   The surgical perspectives were described so in detail and the intra-operative images were included. These may not be necessary unless the title of this manuscript was modified to emphasize the surgical aspect.

l   The patient's basic data, including physical finding, pancreatic enzyme levels (amylase/lipase), were inadequately described.

l   Images of one series were included in different figures, such as Fig 2 and Fig 3, which belong to a single MRI series. Fig 7~10 belong to one series of operative images.

l   Overall, although the article provides valuable insights into the diagnosis and management of pancreatic duct lithiasis, more attention to clarity, grammar, consistency, and conciseness could enhance its readability and impact.

Author Response

Please see the attachment below.

Reviewer 4 Report

Comments and Suggestions for Authors

This is a case report discussing mainly the surgical management of cases with multiple pancreatic duct stones. There are many spelling mistakes and the authors get lost in their titles. Many pieces of information are under the wrong titles and many sections are too long. It could not be published in such shape. I just will give some guidance below.

Introduction:

Too long with multiple headings (from line to line 200). All these had to be merged into a single title (introduction) in a maximum of one page.

Line 61: and so on----> to be removed.

Line 200: Materials and methods------> to be removed and add title Case report.

Line 201-205: A 60-year-old patient with a history of grade III essential hypertension with high cardiovascular risk, perforated gastric ulcer (Pean-Billroth I Gastrectomy), chronic pancreatitis caused by ethanol (Rosemont classification), chronic smoking, presents himself in emergency conditions accusing abdominal pain of moderate intensity, with epigastric localization and posterior irradiation, respectively dyspepsia------------> very bad case presentation. to be rephrased.

Line 244: Results----> to be included in the title case report.

Line 250: Expected Outcome of the Treatment Plan-----> to be removed and added to the review section.

Line 282: Follow-up is essential in pancreatic lithiasis for regular assessment of clinical status, monitoring of biological parameters, and imaging evaluation. Ideally, post-therapeutic homeostasis should be achieved, characterised by normalisation of organ-specific biologcal parameters------------->to be removed.

Line 312: the part of the patient care and procedure to be removed and added to the section of the case report.

Comments on the Quality of English Language

Very bad and extensive editing of the English language is required.

Author Response

Please see the attachment below.

Reviewer 5 Report

Comments and Suggestions for Authors

In this study, Dan Brebu et al report a case of chronic pancreatitis receiving spleen sparing distal pancreatectomy. It is an interesting case and I have some comments for this manuscript:

1.      Although the abstract is succinct and concise, the section of introduction is redundant for the type of case report. Could authors give pictures of pancreatic anatomy with vital structures, such as vessels. It is difficult to read the texts without pictures to understand the anatomy of pancreatic structures.

2.      As authors’ mention, “the main cause in the etiopathogenesis of chronic pancreatitis is represented by chronic alcoholism (75%)” (line 137). But alcoholism could not be seen in the Figure 1 (Etiopathogenesis of chronic pancreatitis…). Moreover, how about the roles of autoimmune or IgG4 pancreatitis in the etiopathogenesis of chronic pancreatitis? What is the meaning of the last item of “acute pancreatitis” in the figure 1? Dose it mean “recurrent acute pancreatitis”?

3.      In the section of material and methods, authors report the laboratory information of this case. But some values were missed, such as amylase/lipase, direct or total bilirubin, alkaline phosphatase, gamma glutamyl transpeptidase, triglyceride, calcium for the differentiated diagnosis of acute cholangitis or cholecystitis. What are the meanings of leukocytosis (15.7/%) and cholinesterase 5822 U/L? (line 207, 208)

4.      Some abbreviations should be explained, such as CBIH, CBP (line 210, 211). What is the meaning of “distended gallbladder 10.8/3.2 cm”? (line 211)

5.      The section of result may be revised. Some information, such as literature review, of this section should be presented in the section of discussion. The texts in line 217-224 and line 230-232 are duplicated.

6.      Figure 7-9 is difficult to be understood for those not in the field of general surgery. Could authors give some descriptions, such as arrow/head in these pictures, or hand drawing pictures?

Author Response

Please see the attachment below.

Round 2

Reviewer 3 Report

Comments and Suggestions for Authors

The manuscript format was greatly revised and the title was revised accordingly to the content.

However, further revisions were still needed and some questions is to be answered.

1.      I wondered why pancreatic enzyme data (amylase and lipase) was not provided in the patient presentation since acute on chronic disease was suspected.

2.      The data of cholinesterase was included and was quoted to be one parameter of Ranson’s criteria with AST/LDH. However, by the definition of Ranson’s criteria, cholinesterase is not one of the parameters. The role of cholinesterase in this presentation is doubtful.  

3.      Please explained why some descriptions in the discussion section was quoted by ([name]) rather than by the Reference No.

4.      Some grammar and spellings should still be revised: (leukocytosis 15.7/%-->?)

Author Response

Please see the attachment below.

Reviewer 4 Report

Comments and Suggestions for Authors It is in better shape and can be published in the present form.

Comments on the Quality of English Language It is in better shape and can be published in the present form.

Author Response

Please see the attachment below. 
